# Assessment of decadal changes in coastal nitrogen sources in NW Spain with stable isotopes in macroalgae and mussels

Antonio Bode[1]*, Rita García-Seoane[1], Zulema Varela[2], Inés G. Viana[1]

1 Centro Oceanográfico de A Coruña, Instituto Español de Oceanografía (IEO-CSIC), A Coruña, Spain,
2 CRETUS, Ecology Unit, Department of Functional Biology, Universidade de Santiago de Compostela, Santiago de Compostela, Spain

* antonio.bode@ieo.csic.es

## Abstract

Upwelling is one of the major mechanisms responsible for the input of nutrients sustaining high levels of marine primary production. As a consequence of global change, variations in upwelling intensity may affect nutrient supply thus impacting marine food webs. In this study, we examine the effects of decadal variability of upwelling strength on nitrate supply and its influence on the nitrogen stable isotope composition at the base of the marine food web at the northern boundary of the Canary Current upwelling system (NW Spain) between 1989 and 2023. The study focused on the early upwelling season each year (March-June) to minimize the effects of nitrogen remineralization. Intertidal macroalgae (Phaeophyceae) and mussels (*Mytilus galloprovincialis*) were used as proxies for temporally integrated isotopic signals of nitrogen sources ($\delta^{15}N$). While no significant temporal trends for either upwelling strength, nitrate concentrations, or stable isotopes were found, three periods with characteristic upwelling and nutrient regimes were identified. A linear increase in $\delta^{15}N$, particularly in *Fucus* spp., associated with a decreasing contribution of upwelling-derived nitrogen suggest the influence of additional sources, likely of anthropogenic origin. Thus, no net change in productivity would be expected in this region despite quasi-decadal shifts in upwelling dynamics. Further insights on the origin and relevance of these sources can be gained through the investigation of river and runoff inputs and the use of more sensitive tracers, such as amino acid $\delta^{15}N$ analysis in mussels.

## Introduction

Nitrogen availability is one of the major elements regulating primary production in the ocean [1]. While nitrogen can be regenerated *in situ* from organic substrates, the input of inorganic nitrogen from external sources (new nitrogen) allows for the net increase in primary production, its transfer to upper trophic levels, and the export to sediments

**Data availability statement:** Upwelling index values are available at http://www.indicedeaflo-ramiento.ieo.es. Hydrographic data are available at https://doi.org/10.1594/PANGAEA.919087 and at https://cdi.seadatanet.org/csr/ (project RADIALES). Stable isotope data are available at https://doi.org/10.1594/PANGAEA.974186 and https://doi.org/10.1594/PANGAEA.974299

**Funding:** This research was supported by Agencia Estatal de Investigación (MCIN/AEI/10.13039/501100011033 Spain) through project QLOCKS (PID2020-115620RB-100), and additional funds by the Consellería de Medio Ambiente (Galicia, Spain) through the 3rd phase of the Environmental Specimen Bank (2000–2004), the Competitive Reference Group GRC GI-1252/GPC2020-23 (ED431C 2020/19), co-funded by the Xunta de Galicia (Spain) and the European Regional Development Fund (ERDF, EU), and postdoctoral research grants Juan de la Cierva-Incorporación to I.G.V. (IJC2019-040554-I), Juan de la Cierva-Formación to R.G.S. (FJC2019-040921-I), and María Zambrano Programme of the Spanish Ministry of Universities to Z.V. R.G.S also received funds from the European Union NextGenerationEU/PRTR programmes and is currently supported by the Horizon Europe research and innovation programme under a Marie Skłodowska-Curie Postdoctoral Fellowship 2023 (101150001-PelCon). The funders had no role in study design, data collection and analysis, decision to publish, or preparation of the manuscript.

**Competing interests:** The authors have declared that no competing interests exist.

[2]. The latter process is of foremost importance for the burial of organic matter and the consequent removal of carbon over long time scales [3]. New nitrogen from deep waters, mostly in the form of nitrate, is delivered to the illuminated surface layer by upwelling, eddies, winter mixing, diffusion across the pycnocline [4], diazotrophic bacteria [5], and as atmospheric wet deposition [6,7]. Rivers can also supply considerable amounts of nitrogen but generally with limited spatial influence [8,9]. Diazotrophy, eddies and diffusion are the main sources of new nitrogen in the permanently stratified waters covering most of the ocean surface, generally resulting in low values of net primary production [10,11]. In contrast, upwelling [5,12,13] and mixing caused by the cooling of surface waters during winter [14] provide large amounts of new nitrogen leading to excess primary production and biomass accumulations.

An increase in wind-driven upwelling at the eastern boundary of continents has been predicted as a consequence of global warming [15–17]. The main mechanism enhancing this phenomenon would be a strengthened temperature gradient between land and ocean causing an intensification of alongshore winds and hence, the input of new nitrogen to the ocean surface waters. However, biogeochemical models suggest that the response of upwelling systems would be highly variable depending on factors such as deep-water ventilation [18] and geographical location, the latter because the effect of the poleward displacement of high-pressure cells on wind fields may outweigh over the effect of land-ocean temperature differences [19]. Analysis of wind series observations [16,17,20] also revealed heterogeneous spatial responses in the wind regime across major upwelling systems. Similarly, the long-term response of the biological communities to increased nutrient inputs appears to be complex, suggesting changes in food web dynamics [21–23].

Nitrogen sources can be traced into the food web using stable isotopes, as the isotopic composition of nitrogen in a given substrate is influenced by the biogeochemical processes and the environmental conditions involved in its formation [24–26]. For instance, nitrate produced by nitrification is typically more depleted in the heavier isotope ($^{15}N$) than nitrate remaining after denitrification or phytoplankton assimilation processes [24]. Comparatively, nitrate from anthropogenic sources is highly depleted (e.g., synthetic fertilizers) or highly enriched (e.g., manure and septic waste) in $^{15}N$ [27]. Thus, these sources can be traced by analyzing the isotopic composition of either dissolved nitrate [28,29] or the tissues of organisms growing in the ecosystems of interest [30–32]. While the former is a direct measurement of the instantaneous composition of seawater, the latter is a time-integrated measurement of the nitrogen assimilated by the organism. The apical segment of perennial macroalgae (e.g., some Fucaceae) can integrate the isotopic composition of ambient nitrogen at time scales of several weeks [33,34], whereas the muscle of some filter feeders, such as mussels, integrates the isotopes over several months [35].

In NW Spain, at the northern limit of the Canary Current Eastern Boundary Upwelling (NW Spain), the seasonal influence of coastal upwelling is considered the major source of nitrogen for new production [12,36]. Upwelling of the nitrate-rich Eastern North Atlantic Central Water (ENACW) typically occurs mainly between March and October on the western region (Galicia) and its influence decreases in the northern

region (mar Cantábrico), where its effects are generally restricted to the summer season and to the westernmost area [37,38]. The presence of rías in Galicia further enhances the effect of upwelling by facilitating the use of recycled nutrients to primary producers [12]. In the Rías Baixas zone, south of 43°N, the orientation and depth of the rías favors the penetration of upwelling nutrients and its renewal by remineralization. Conversely, the input of upwelling nutrients in northern rías (Rías Altas) is restricted by their shallower depth and the presence of nutrient-poor shelf waters [39]. The dynamics of the upwelling suggested changes in the nutrient enrichment and biological productivity al decadal time scales [12,40] although a most recent examination of nutrient time series revealed variable trends depending on the length of the time period considered, suggesting increasing nutrient inputs from continental sources [22,41,42]. Previous studies in this region showed a clear influence of upwelling-derived nitrogen in the isotopic composition of plankton during spring and early summer, while the impact was less evident in other seasons likely due to remineralization and nitrogen inputs from other sources [43,44]. In turn, stable isotopes in intertidal macrophytes tracked the geographic variability in upwelling along the coast [45] but pointed out to a large prevalence of anthropogenic nutrients, at least in sheltered areas inside the rías [45–49]. However, the potential contribution of new nitrogen from anthropogenic sources in this region remains poorly constrained in space and time.

In this study, we analyzed decadal changes in new nitrogen inputs to the coastal ecosystem in NW Spain by examining historical and new data of stable nitrogen isotopes in intertidal macroalgae and mussels collected between 1989 and 2023. We focused on three zones, with reportedly differential effects of upwelling: Rías Baixas (south of Cape Finisterre at 43°N), Rías Altas (north of Cape Finisterre and west of 8°W), and mar Cantábrico (east of 8°W).

## Materials and methods

### Environmental data

Upwelling intensity was determined using Ekman transport values estimated from surface wind data [50], as reported in the form of the Upwelling Index (UI) by the Instituto Español de Oceanografía (http://www.indicedeafloramiento.ieo.es). First, long term trends were examined using monthly values of UI for the period 1989–2023 (S1 Fig). Then, for each year, UI values corresponding to the period of maximum growth of algae (March to June, [51]) were averaged within three 1° x 1° degree cells representative of the study coastal zones (Fig 1). Positive values of this index indicate net upwelling

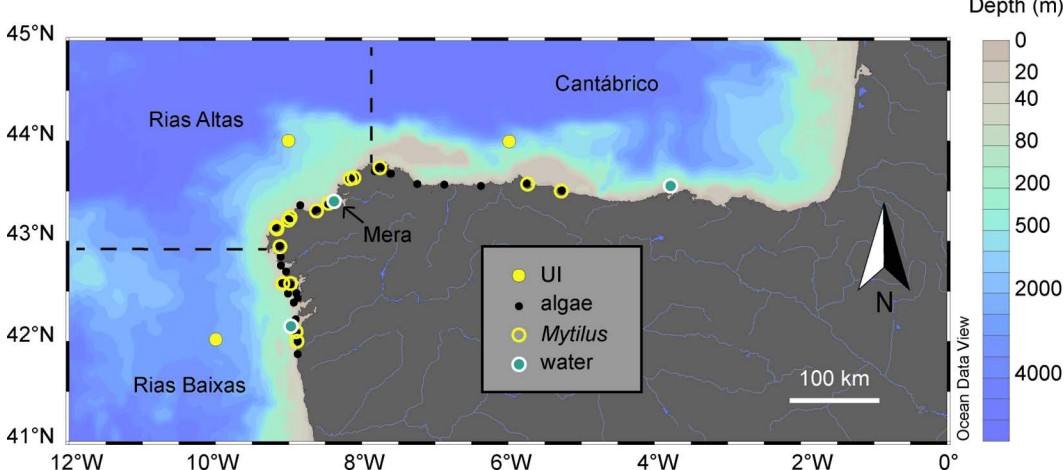

**Fig 1. Sampling locations.** Map of sampling locations for macroalgae (black dots), mussels (*Mytilus*, yellow circles) and water (blue dots), along with the position of grid centers for the computation of upwelling index values (UI, yellow dots). The separation between zones (Rías Baixas, Rías Altas, and Cantábrico) is indicated by dashed lines, and the bathymetry (Depth, m) by a blue-tan gradient color scale. The arrow marks the study site (Mera) for the analysis of seasonal variability in δ15N. Map provided by Ocean Data View (Schlitzer, R., Ocean Data View, https://odv.awi.de, 2025).

periods when surface water is transported offshore while negative values indicate an accumulation of surface water against the coast (downwelling).

Temperature, salinity and nitrate concentrations were obtained from the observational program RADIALES [52]. One shelf station with a maximum depth of 100 m was selected at each zone (Fig 1). Profiles of temperature and salinity were obtained with a CTD (Seabird SBE25), and nitrate concentrations were determined by colorimetric analysis of frozen water samples preserved in polyethylene vials [22]. In this study, we used only surface (0–5 m) and near bottom (70–100 m) values of these variables averaged from March to June, as for UI. In addition, the input of new nitrate from ENACW was estimated from the observed water temperature near the bottom of each station using the equations in Álvarez-Salgado et al. [12]. The difference between the estimated and observed nitrate concentrations (nitrate anomaly, $\Delta NO_3$) was used as an index of the relative importance of upwelling compared to other nitrate sources. This index would be equal to zero when the measured nitrate concentration matched the average concentration expected in ENACW at a given temperature. Negative values of $\Delta NO_3$ point to a deficit of ENACW nitrate due to a reduced upwelling strength or to the partial consumption of upwelled nitrate before reaching the coast. Finally, positive $\Delta NO_3$ values indicate the input of nitrate from other sources in addition to ENACW.

## Stable isotopes in intertidal sentinels

Species of brown algae (Phaeophyceae) and mussels (*Mytilus galloprovincialis*) were selected as biomonitors of nitrogen utilization in the coastal food web. To ensure maximum geographical and temporal coverage across the selected zones (Fig 1), several species of *Fucus* were considered (*F. vesiculosus*, *F. spiralis*, and *F. serratus*). When these species were not present (13% of cases) *Bifurcaria bifurcata* was used as a substitute. The latter is a warm-water species gradually replacing *Fucus* spp. in the southern Bay of Biscay [53] and contributes to nitrogen transfer in littoral food webs [54]. Cross-species calibrations were made to ensure the consistency of the reconstructed isotopic time series. Historical data of nitrogen isotopic measurements in algae were extracted from the Environmental Specimen Bank of Galicia time series (ESBG, [46]) and from past studies at the Instituto Español de Oceanografía [45,48,54,55]. Only data from specimens collected at locations under the direct influence of upwelling waters were considered, as the aim was to reconstruct long-term trends in stable isotopes and upwelling variations. Additional samples of algae and mussels were collected in summer 2023 to extend the time series duration. Collection of intertidal specimens was conducted under permits granted by Xunta de Galicia, Spain (Consellería de Medio Ambiente, permit Environmental Specimen Bank project, and Consellería do Mar, permit 22062010 ANILE project). Both algae and mussels were collected at mid to low intertidal levels during low tide at wave-exposed locations outside rías and estuaries, avoiding locations near waste-water treatment plants and large urban settlements, thus maximizing the probability of detecting the influence of marine nitrogen sources [45,46,54]. The shell length of the mussels (as a proxy for age) was measured to investigate potential ontogenetic changes of isotopic composition [56].

The distal 3 cm of the algal thallus and portions of the adductor muscle of mussels were oven dried (45–50°C, 48 h) or freeze-dried (2023 samples) before grinding and preparing for isotopic determinations. Samples were stored in dry containers for up to 6 months before analysis. Based on estimated isotopic turnover rates in these organisms, the isotopic composition of nitrogen in the apical segments of algae and the muscular tissue of mussels represent the integrated assimilation of nitrogen over the previous 3 months [34] and up to 10 months [57], respectively. Individual specimens were used for isotope analysis except for samples acquired from the ESBG collection (each sample was a pool of up to 30 individuals).

Determinations of the natural abundance of stable nitrogen isotopes were made at the Servicio de Apoyo a la Investigación (SAI) of the Universidad de A Coruña (Spain), using an isotope-ratio mass spectrometer coupled to an elemental analyzer, as described in previous publications [45,46,54]. Isotopic abundance was expressed as $\delta^{15}N$ (‰) relative to atmospheric nitrogen [58] by using standards (IAEA-N-1, IAEA-N-2, and IAEA-NO-3) of the International Atomic Energy

Agency (Vienna, Austria). Precision (± se) of triplicate $\delta^{15}N$ determinations of standards and samples was < 0.03 ‰. In addition, $\delta^{15}N$ in amino acids (i.e., alanine, glycine, threonine, serine, valine, leucine, isoleucine, proline, aspartic acid + asparagine, methionine, glutamic acid + glutamine, phenylalanine, and lysine) was analyzed in a set of mussel samples collected in 2023 by gas chromatography coupled to isotope-ratio mass spectrometry as described in McCarthy et al. [59]. Precision (± se) for $\delta^{15}N$ of individual amino acids was < 1‰. In this study we use only the values for phenylalanine ($\delta^{15}N_{Phe}$) as an indicator of nitrogen source for primary producers [60]. All isotopic measurements considered in this study (including the amino acid $\delta^{15}N$ values in mussels) are available at the PANGAEA repository (http://www.pangaea.de, [61,62]).

## Statistical analysis

Long-term series of environmental data (UI, temperature, salinity, nitrate) were statistically assessed for significant trends using the Mann-Kendall test [63]. In addition, multiannual upwelling regimes were identified using the accumulated anomalies method [64]. Briefly, the cumulative sum of residuals of each observation differing from the time series mean (ΔUI) was plotted to detect anomalies in slope, these being indicative of upwelling regimes with values higher (positive slope) or lower (negative slope) than the overall mean. This method was previously applied to a subset of the UI series in the study region [22]. Significance of differences in mean values of environmental and isotopic variables between the identified UI regimes were assessed using ANOVA and Duncan tests after data exploration following Zuur et al. [65]. Comparisons and calibration of isotopic variables among species (algae) or seasons (mussels), and the relationships between the $\delta^{15}N$ and $\Delta NO_3$ were made by means of Student-$t$ tests and linear regressions. Non-linear regressions were applied to correct for seasonal variation in the isotopic signal of algae. All statistical analyses were computed using PAST v4.09 software [66].

## Results

### Upwelling and environmental variables

The long-term series of UI values averaged from March to June for each year did not show significant long-term trends in the studied period (Mann Kendall test, $P > 0.05$) for any of the zones considered (Fig 2a). However, the accumulated UI anomalies enabled the identification of three distinct regimes of upwelling dynamics affecting all zones (Fig 2b). Between 1989 and 1997, and between 2009 and 2023, UI reached higher values than the overall mean, particularly in Rías Altas ($F_{2,33} = 4.233$, $P < 0.05$) and mar Cantábrico zones ($F_{2,33} = 7.149$, $P < 0.01$), while between 1998 and 2008 UI values were in general lower than the series mean (Table 1). Notwithstanding the general synchrony between zones, no significant differences in mean UI were observed between periods in the Rías Baixas zone ($F_{2,33} = 1.779$, $P > 0.05$).

Environmental variables were averaged between March and June and $\delta^{15}N$ (‰) was adjusted for the seasonal minimum for *Fucus* spp. in each year. Standard deviation (sd), UI: upwelling index ($m^3$ $s^{-1}$ $km^{-1}$), SBT: bottom temperature (°C), $BNO_3$: bottom nitrate (µM), $\Delta NO_3$: difference (µM) between $BNO_3$ and nitrate concentration in Eastern North Atlantic Central Waters computed from equations in Álvarez-Salgado *et al.* [12]. Significant differences between period means (ANOVA and *post-hoc* Duncan tests, $P < 0.05$) for each variable and zone are indicated by different letters (group). N: number of data.

Similarly, because of large year to year variability (S2 Fig), long term trends were not significant for any variable considered (Mann Kendall test, $P > 0.05$). Moreover, mean values of environmental variables did not differ significantly between zones across the three recognized upwelling regimes, except for $\Delta NO_3$ (Table 1). The latter showed significantly lower values in the Rías Baixas zone during the 1998–2008 regime when compared to the previous and subsequent regimes ($F_{2,27} = 8.824$, $P < 0.01$), revealing a relative deficit in upwelling nitrate in bottom waters in this zone (Fig 3, Table 1). It must be noted that most $\Delta NO_3$ values were negative, particularly in the Rías Altas and Cantábrico zones, thus reflecting the decreasing influence of upwelling northwards.

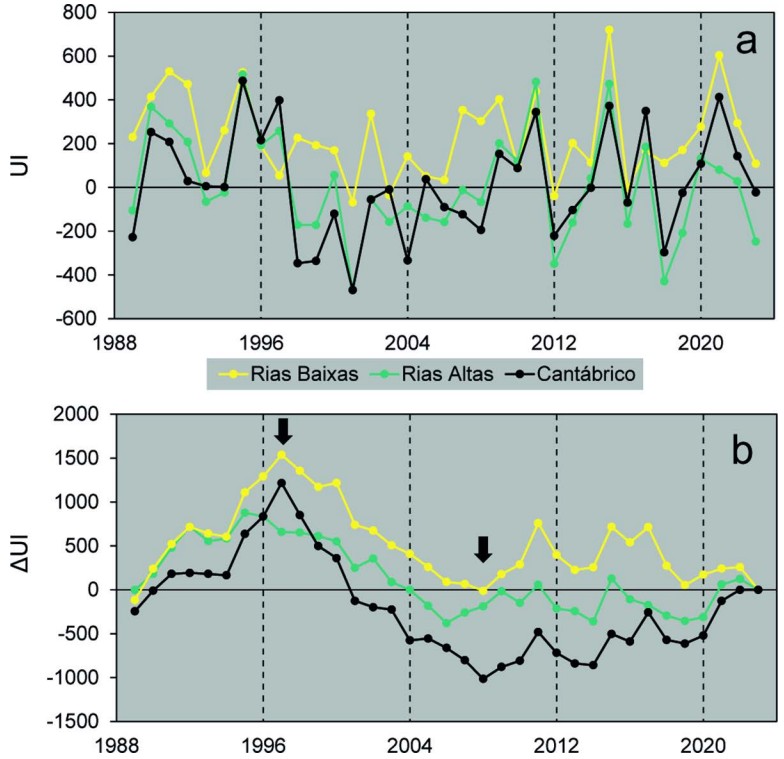

**Fig 2. Upwelling index series.** Values of (a) the upwelling index (UI, m³ s⁻¹ km⁻¹) and (b) the accumulated anomalies from the series mean (ΔUI) averaged between March and June for the three zones considered (see Fig 1). The arrows indicate the inflection points between periods of increasing or decreasing anomalies.

## δ¹⁵N in macroalgae

Because of the difficulties in finding the same algal bioindicator at all zones and periods, isotopic signal measured in different species were calibrated to obtain comparable δ¹⁵N values across zones and years. First, δ¹⁵N measured in all species of *Fucus* were compared when collected concurrently. As the mean δ¹⁵N values for *F. spiralis* and *F. vesiculosus* were not significantly different (S3 Fig, Student-$t$ = 0.392, $P$ > 0.05) values for these species were employed upon local availability and subsequently labelled as *Fucus* spp. In turn, values for *F. serratus* were converted to their equivalents in *Fucus* spp. using a linear regression (S4a Fig). Similarly, δ¹⁵N of *B. bifurcata* followed a linear relationship with δ¹⁵N of *Fucus* spp. (S4b Fig).

Seasonality was a major factor of variability in δ¹⁵N of *Fucus* spp. Consequently, a non-linear function fitted to the δ¹⁵N deviations from the annual minimum for *F. spiralis* sampled at the same location across multiple years (S5 Fig) was used to correct the raw δ¹⁵N values obtained from algae sampled at different seasons. The fitted annual minimum occurred on Julian day 195 (i.e., mid-July) and was assumed to represent the maximum influence of upwelling nitrate and a relatively low influence of isotopically enriched sources.

Even when most of the observations concentrated in the central period of the whole series, the final series of δ¹⁵N for *Fucus* spp., once adjusted for seasonality (δ¹⁵N$_{Fucus}$), revealed relatively large variations in annual means in the Cantábrico when compared to the other zones (Fig 4 a, c, e). Regarding differences between UI periods, mean δ¹⁵N$_{Fucus}$ was significantly lower in 1998–2008 in the Rías Baixas ($F_{2,119}$ = 5.942, $P$ < 0.01) and Rías Altas ($F_{2,56}$ = 15.370, $P$ < 0.001) zones when compared to mean values in earlier and later stages (Table 1). In contrast, the lowest δ¹⁵N$_{Fucus}$ in the Cantábrico zone was found for the 2009–2023 period ($F_{2,62}$ = 9.858, $P$ < 0.001).

Table 1. Environmental variables and δ15N of *Fucus* spp. and *Mytilus galloprovincialis* averaged for sampling zones and upwelling periods.

| variable | zone | 1989-1997 | | | | 1998-2008 | | | | 2009-2023 | | | |
|---|---|---|---|---|---|---|---|---|---|---|---|---|---|
| | | mean | sd | N | group | mean | sd | N | group | mean | sd | N | group |
| UI | Rías Baixas | 306.87 | 176.23 | 10 | a | 155.09 | 145.78 | 11 | a | 244.54 | 215.57 | 15 | a |
| | Rías Altas | 161.02 | 209.00 | 10 | a | −129.20 | 133.66 | 11 | b | 12.40 | 271.83 | 15 | a,b |
| | Cantábrico | 119.19 | 234.16 | 10 | a | −185.25 | 163.13 | 11 | b | 82.35 | 217.87 | 15 | a |
| SBT | Rías Baixas | 12.79 | 0.56 | 10 | a | 12.62 | 0.42 | 11 | a | 12.70 | 0.47 | 15 | a |
| | Rías Altas | 12.92 | 0.69 | 10 | a | 12.93 | 0.30 | 11 | a | 12.89 | 0.35 | 15 | a |
| | Cantábrico | 12.26 | 0.13 | 10 | a | 12.25 | 0.40 | 11 | a | 12.20 | 0.27 | 15 | a |
| BNO$_3$ | Rías Baixas | 7.95 | 2.85 | 4 | a | 7.96 | 1.86 | 11 | a | 7.18 | 1.77 | 15 | a |
| | Rías Altas | 5.39 | 2.02 | 10 | a | 5.49 | 1.40 | 11 | a | 5.73 | 1.24 | 15 | a |
| | Cantábrico | 6.61 | 2.09 | 4 | a | 5.70 | 1.65 | 11 | a | 6.35 | 1.62 | 15 | a |
| ΔNO$_3$ | Rías Baixas | 1.35 | 1.56 | 4 | a | −2.54 | 2.16 | 11 | b | 0.11 | 1.77 | 15 | a |
| | Rías Altas | −0.86 | 1.09 | 10 | a | −0.60 | 1.60 | 11 | a | −0.64 | 1.52 | 15 | a |
| | Cantábrico | −1.63 | 1.96 | 4 | a | −2.73 | 1.68 | 11 | a | −2.37 | 1.52 | 11 | a |
| δ15N$_{Fucus}$ | Rías Baixas | 6.38 | 0.23 | 4 | a | 5.51 | 0.48 | 30 | b | 6.15 | 1.01 | 25 | a |
| | Rías Altas | 5.90 | 0.15 | 3 | a,b | 5.35 | 1.55 | 45 | b | 6.61 | 0.95 | 74 | a |
| | Cantábrico | 7.03 | 0.40 | 3 | a | 5.96 | 1.49 | 30 | a | 4.84 | 0.80 | 32 | b |
| δ15N$_{Mytilus}$ | Rías Baixas | --- | --- | --- | --- | 7.88 | 0.61 | 28 | a | 7.40 | 0.54 | 21 | b |
| | Rías Altas | --- | --- | --- | --- | 7.57 | 0.85 | 166 | a | 6.92 | 1.28 | 47 | b |
| | Cantábrico | --- | --- | --- | --- | --- | --- | --- | --- | 6.71 | 0.62 | 22 | --- |

## δ15N in mussels

In the case of *M. galloprovincialis*, it was difficult to obtain representative values in the historical archive because δ15N increased with mussel size (S6 Fig). Therefore, only values obtained from individuals larger than 25 mm in shell length were included in the time series analysis. In contrast to *Fucus* spp., δ15N anomalies from the annual minimum in mussels followed a linear relationship with time along the year (S7 Fig), with minimum δ15N values typically observed in winter. However, for comparison with the environmental and *Fucus* spp. series, mussel δ15N (δ15N$_{Mytilus}$) were adjusted for the date of the annual minimum in *Fucus* spp. (i.e., Julian day 195).

The reconstructed mussel δ15N series partially covered the considered UI periods and zones, showing comparatively less variability than those of *Fucus* spp. (Fig 4 b, d, f). Nevertheless, mean δ15N values for the 1998–2008 UI period were significantly higher than those measured in the 2009–2023 period for both Rías Altas ($F_{1,211} = 16.545$, $P < 0.001$) and Rías Baixas ($F_{1,47} = 18.182$, $P < 0.001$) zones (Table 1). There were not enough data to investigate temporal differences for the Cantábrico zone although mean values for the 2009–2023 period were similar to those of the Rías Altas zone, suggesting a similar pattern.

## Relationships between δ15N and ΔNO$_3$

Annual means of δ15N for *Fucus* spp. increased linearly with those of ΔNO$_3$ when considering all zones ($r = 0.417$, $P < 0.05$; Fig 5a). A similar pattern was observed with mussel δ15N, although the correlation was not significant likely due to the small sample size (Fig 5b). Further investigation of this relationship using samples from 2023 revealed significantly higher mean values of δ15N$_{Phe}$ ($F_{2,15} = 4.263$, $P < 0.05$) and ΔNO$_3$ ($F_{2,9} = 10,500$, $P < 0.001$) for the Rías Altas zone, while bulk mussel δ15N revealed a different variability pattern among zones (Table 2). It is important to note that other environmental variables (i.e., UI, bottom temperature, and nitrate) did not show significant differences among zones in 2023.

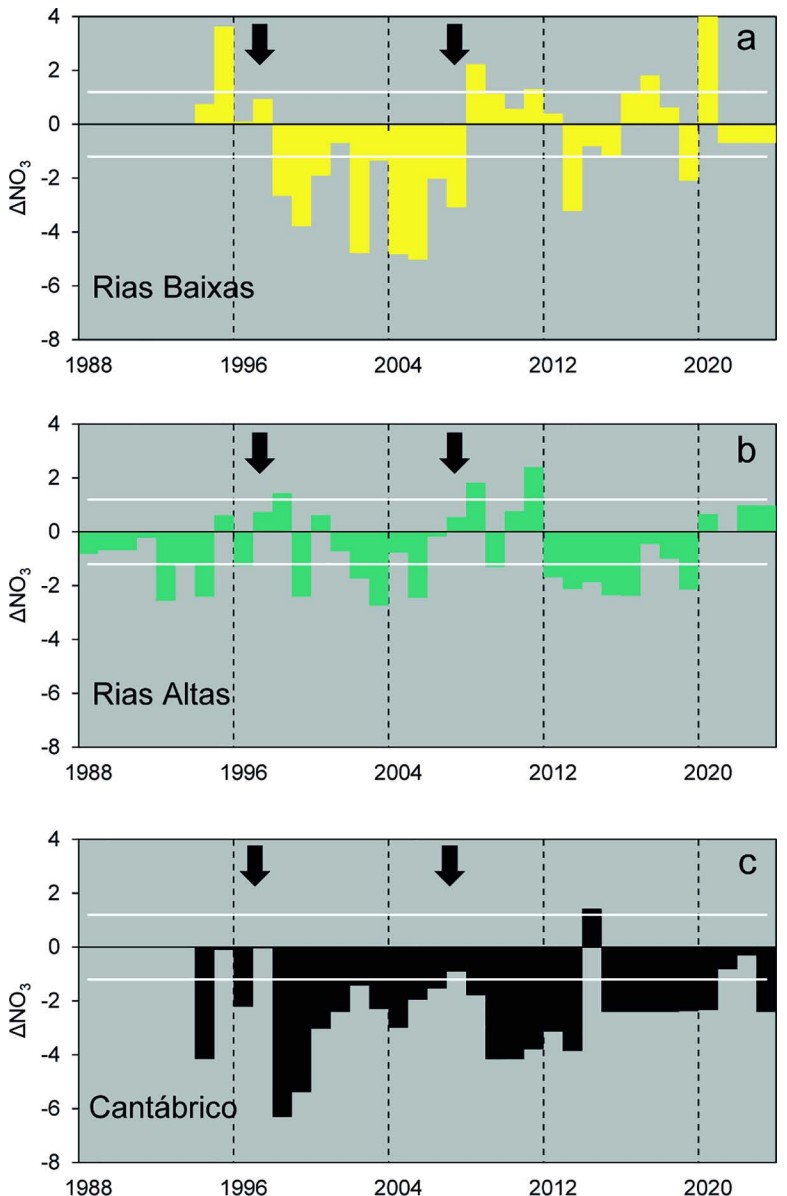

**Fig 3. Nitrate anomaly series.** Time series of anomalies of nitrate (ΔNO3, μM) from concentrations expected in Eastern North Atlantic Central Waters (ENACW) in bottom water (70-100 m) averaged from March to June for Rías Baixas **(a)**, Rías Altas **(b)**, and Cantábrico **(c)**. The horizontal dashed lines indicate ± 1 standard error of the estimation. Expected ENACW nitrate computed from equations in Alvarez-Salgado et al. [12]. The black arrows indicate the inflection points of the upwelling index series (see Fig 2).

Environmental variables and $\delta^{15}N$ (‰) in *Mytilus galloprovincialis* were averaged for the different zones (see Fig 1) in 2023. Variables are defined in Table 1. $\delta^{15}N_{Phe}$ is the natural abundance of nitrogen isotopes in phenylalanine (‰). Significant differences between zone means (ANOVA and *post-hoc* Duncan tests, $P < 0.05$) are indicated by different letters (group). N: number of data.

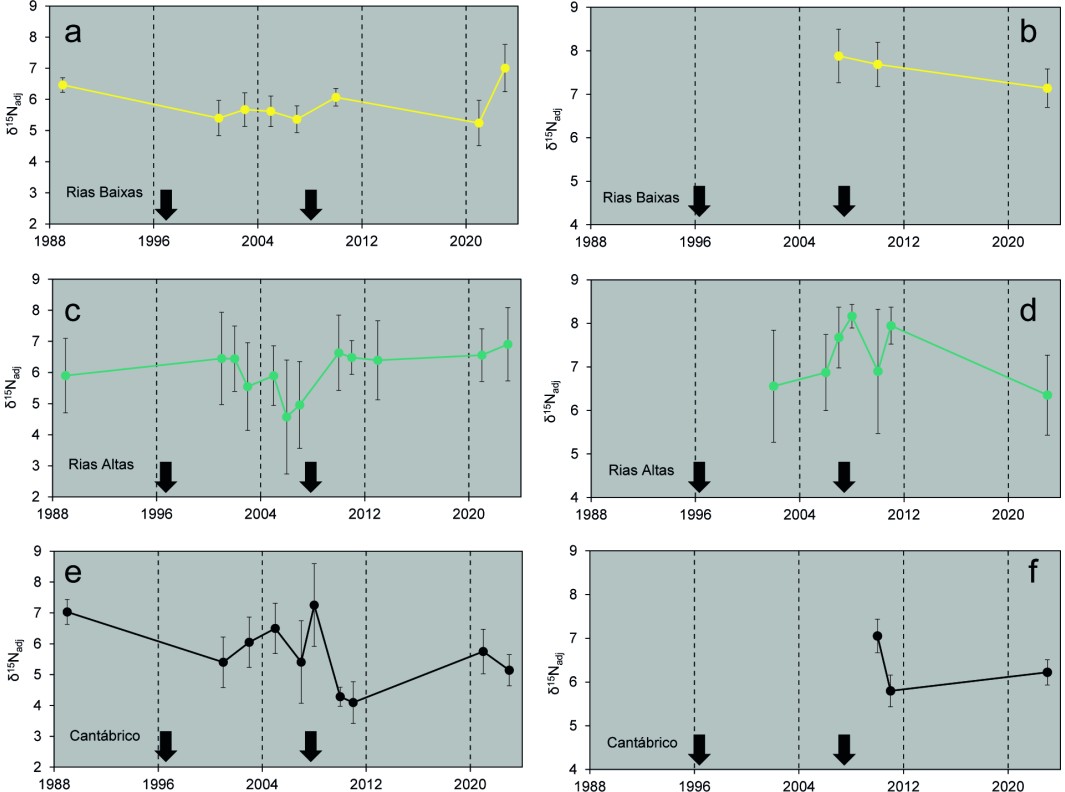

**Fig 4. Stable isotopes in algae and mussels.** Mean (± sd) δ15N values of *Fucus* spp. (a, c, e) or *M. galloprovincialis* (b, d, f) adjusted for the seasonal minimum value for *Fucus* spp. in each year (δ15N$_{adj}$), for the Rías Baixas **(a, b)**, Rías Altas **(c, d)**, and mar Cantábrico zones **(e, f)**. The black arrows indicate the inflection points of the upwelling index series (see Fig 2).

## Discussion

### Changes in upwelling and nutrient enrichment

Our results do not support a generalized intensification of upwelling in recent decades due to global climate change [15], nor a weakening of upwelling in the northern limit of the Canary Current Eastern Boundary Upwelling System associated with a reduction in northerlies [40]. Rather, the findings are in line with the projections of heterogeneous responses of upwelling related to the displacement of high-pressure cells [19]. In this region, spring and early summer upwelling showed distinct periods of predominantly increasing (or decreasing) intensity with quasi-decadal periodicity, as previously reported from a subset of this series considering annual averages [22]. The mean intensities of upwelling, however, were not significant between periods for the Rías Baixas zone, where upwelling is generally stronger. Therefore, no consistent linear trends were evident across the three-decade study period, despite climate models predict a generalized increase in upwelling intensity and frequency during spring and summer towards the end of the 21st century [67]. These changes also imply variability in nutrient inputs following the upwelling periods and geographical location.

Overall, we did not find evidence of systematic trends in nitrogen enrichment or depletion in the region. Instead, our results suggest a compensation of upwelling nutrients with other sources maintaining relatively steady concentrations. This compensatory mechanism was already advanced by previous studies on the rías [40,41]. New nitrogen from the upwelling showed consistent geographical variability, as zero or positive values of ΔNO$_3$ were more frequent in the Rías Baixas than in the Rías Altas and the mar Cantábrico zones. Nevertheless, negative values of ΔNO$_3$, suggesting a deficit

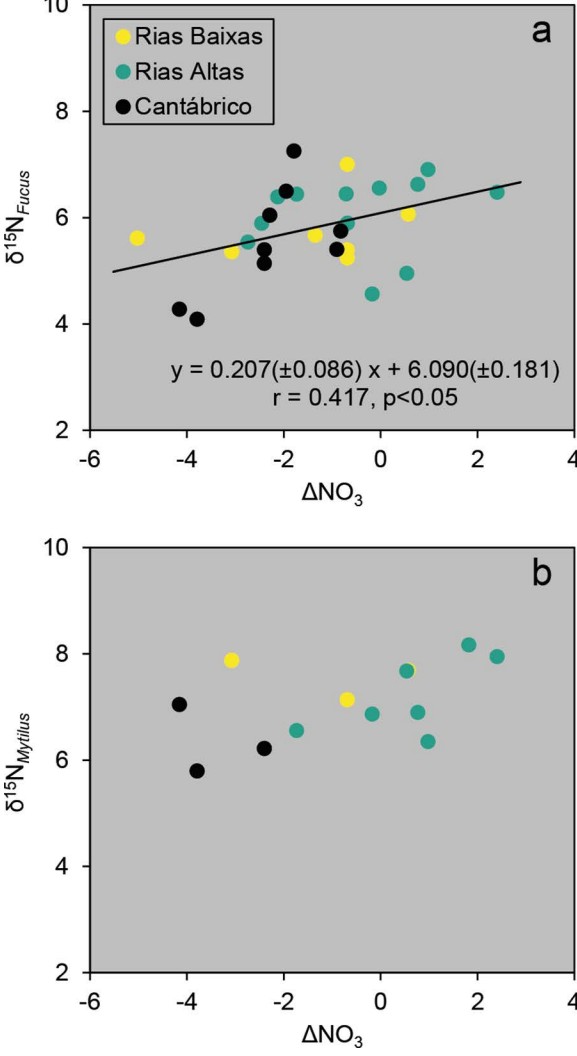

**Fig 5. Relationships between stable isotopes and nitrate anomalies.** Relationships between mean δ15N values (‰) of *Fucus* spp. (a) or *M. gallopro-vincialis* (b) and the mean difference between NACW and observed nitrate concentrations in bottom water ($\Delta NO_3$, μM) from March to June in the three zones indicated in Fig 1. The δ15N values were adjusted for the seasonal minimum value for *Fucus* spp. in each year and zone.

in ENACW nitrate, dominated in all zones and periods, even in the Rías Baixas between 1998 and 2008. These negative anomalies imply a reduced input of nitrate from upwelled ENACW, particularly in the mar Cantábrico, and challenge the current paradigm of the upwelling as the major contributor to nutrient fertilization in this region [36,40]. While nutrient remineralization in Rías and shelf waters has been shown to amplify the initial input of ENACW nitrate on biological productivity [12], this effect is mainly expected in the Rías Baixas [39] and other new nitrogen sources may be of larger importance for the Rías Altas and mar Cantábrico zones. Atmospheric nitrogen fixation and diffusive nitrate inputs through the nitracline have been reported in the study area, particularly during upwelling relaxation events, but they represent a minor fraction of new nitrogen for primary production when compared to nitrate provided by advection [5]. Similarly, riverine nitrate inputs are of locally relevant [68], yet their overall influence on shelf waters appears limited, given the small size of most rivers in the region [9]. Other potential inputs recognized to affect inshore areas inside the rías include nitrogen

**Table 2. Environmental variables and δ¹⁵N of *M. galloprovincialis* for the different zones in 2023.**

| variable | zone | mean | sd | N | group |
|---|---|---|---|---|---|
| UI | Rías Baixas | 108.31 | 816.97 | 4 | a |
| | Rías Altas | −247.44 | 1137.03 | 4 | a |
| | Cantábrico | −22.38 | 827.30 | 4 | a |
| SBT | Rías Baixas | 12.70 | 0.35 | 4 | a |
| | Rías Altas | 13.07 | 0.56 | 4 | a |
| | Cantábrico | 12.24 | 0.08 | 4 | a |
| $BNO_3$ | Rías Baixas | 7.57 | 2.12 | 4 | a |
| | Rías Altas | 6.75 | 3.54 | 4 | a |
| | Cantábrico | 6.15 | 2.04 | 4 | a |
| $\Delta NO_3$ | Rías Baixas | −0.69 | 2.85 | 4 | b |
| | Rías Altas | 0.98 | 0.98 | 4 | a |
| | Cantábrico | −2.40 | 1.66 | 4 | b |
| $\delta^{15}N_{Mytilus}$ | Rías Baixas | 7.13 | 0.38 | 6 | a |
| | Rías Altas | 6.28 | 1.00 | 9 | b |
| | Cantábrico | 6.24 | 0.44 | 3 | b |
| $\delta^{15}N_{Phe}$ | Rías Baixas | 4.85 | 0.51 | 6 | b |
| | Rías Altas | 5.78 | 0.89 | 9 | a |
| | Cantábrico | 4.58 | 0.61 | 3 | b |

from urban waste waters [42,69], fish farms [47] and groundwater [70]. In addition, rainwater may be a significant source of new nitrogen, particularly during oligotrophic conditions [71].

### Tracing N sources in coastal organisms

The relative importance of all these potential sources to coastal food web affecting the biomonitors selected in this study can be inferred by comparing the stable isotope composition of sources and organisms. A compilation of mean δ¹⁵N of the potential sources of new nitrate for the study region revealed a large range of values (Table 3). Rainwater was characterized by negative δ¹⁵N values, while atmospheric nitrogen approached to zero [72], in line with observations for the NE Atlantic [73,74]. Low values were also expected from nitrate in synthetic fertilizers while the highest δ¹⁵N values were reported for animal and urban waste waters [48,75]. In turn, while global averages of groundwater were comparable to those of rivers [76], high δ¹⁵N values were reported for a shallow aquifer close to the study region [77]. Seawater nitrate in this region showed intermediate δ¹⁵N values, with a mean of 6.02‰ for bottom shelf waters, close to the value of 4.88‰ reported for ENACW [78]. In turn, surface shelf waters showed a mean value of 8.28‰, reflecting the presence of sources with higher δ¹⁵N than ENACW.

When compared to nitrate sources, δ¹⁵N values in *Fucus* spp. were closer to those in nitrate from seawater than to values in continental, atmospheric or anthropogenic sources. For instance, the mean value for *Fucus* spp. δ¹⁵N in Rías Altas (6.13 ± 1.35‰, n = 120) was almost equivalent to the δ¹⁵N measured in bottom shelf waters in this zone (6.02 ± 1.37‰, n = 24, Table 3). These results suggest the moderate contribution of nitrate from ¹⁵N enriched sources to the coastal seawater and therefore to the growth of macroalgae. A simple two-source mixing model (ENACW and wastewater) based on values in Table 3 estimated a contribution of 89% from ENACW and only 11% from enriched sources, assuming no isotopic fractionation between nitrate and algal tissue. Similarly, the mean δ¹⁵N value for all mussels analyzed (7.42 ± 0.94, n = 284) would represent a 99% incorporation of ENACW nitrate and 1% of enriched sources, in this case considering a

**Table 3. Reference δ¹⁵N values of nitrate from different sources.**

| source | zone | mean | sd | N | reference | observations |
|---|---|---|---|---|---|---|
| seawater | NE Atlantic, Mera, Galicia, Spain[1] | 8.28 | 0.48 | 13 | [45] | coastal, surface layer |
| | NE Atlantic, A Coruña, Galicia, Spain[1] | 6.02 | 1.37 | 24 | [48] | 70 m depth |
| | N Atlantic > 20°N | 4.88 | 1.70 | 182 | [78] | North Atlantic Central Water |
| estuary | Ria do Burgo, Galicia, Spain[1] | 7.26 | 0.51 | 10 | [45] | surface water |
| rivers | global | 7.10 | 3.80 | 2902 | [77] | surface water |
| air | global | 0.00 | --- | --- | [71] | by convention |
| rainwater | global | −9.21 | 3.81 | 199 | [72] | aerosol |
| | NE Atlantic, W Ireland | −1.00 | 3.00 | 12 | [73] | aerosol |
| | N Atlantic, Bermuda | −2.35 | 4.86 | 126 | [74] | wet deposition |
| groundwater | global | 7.70 | 4.50 | 2291 | [76] | shallow aquifers |
| | Aveiro, N Portugal | 10.88 | 4.64 | 16 | [77] | shallow aquifers |
| WWTP outfall | Galicia, Spain[1] | 16.30 | 1.02 | 7 | [48] | average of two coastal sites[2] |
| | mainland Spain | 11.52 | 7.70 | 8 | [75] | sludge and effluent |
| synthetic fertilizer | mainland Spain | 1.64 | 2.79 | 7 | [75] | ammonium-nitrate and urea |
| animal waste | mainland Spain | 16.67 | 13.28 | 6 | [75] | manure and slurries[3] |

Mean values of δ¹⁵N (‰) in nitrate from potential sources relevant for the study region. Standard deviation (sd). WWTP: waste water treatment plant. N: number of data.

[1]within the region considered in this study.

[2]including additional unpublished data.

[3]from swine, cattle, poultry, and sheep.

δ¹⁵N trophic enrichment of 2.45‰ between diet and mussel muscle, equivalent to the mean difference in δ¹⁵N between the muscle and the digestive gland [79]. Extrapolation of these estimations to the study area can be made by using the empirical relationship between *Fucus* δ¹⁵N and $\Delta NO_3$ (Fig 5 a). In this way, when the observed nitrate concentration in bottom waters corresponds exactly with that expected in ENACW (i.e., $\Delta NO_3 = 0$) the contribution of ¹⁵N enriched sources to the net growth of *Fucus* would be 11%, while the range of this contribution from a large deficit (i.e., $\Delta NO_3 = -5$ μM) to a large excess (i.e., $\Delta NO_3 = 3$ μM) would vary between 2% and 16%.

Notwithstanding that these are crude estimations based on mean values, they support the overall importance of upwelling nitrate for maintaining high levels of biological productivity in the study region, consistent with earlier studies [12,36,40]. The contribution of other new nitrogen sources can be considered as occasional, likely of local importance, and possible helping to sustain relatively high stocks of dissolved nitrogen in shelf waters through the year in Galicia [22,23,39] but not in the mar Cantábrico [80]. These results also support that net growth of intertidal organisms, as evidenced by apical tissues in *Fucus* spp. or muscle tissue in mussels, would rely on new nitrogen provided by the upwelling, whereas metabolic maintenance would be granted by regenerated nitrogen forms.

A major contribution of upwelled nitrogen was also reported for other eastern boundary upwelling ecosystems, although differing on the diversity of the sources traced. For instance, mussel δ¹⁵N reflects latitudinal gradients in the proportion of nitrate upwelled from highly denitrified sources (¹⁵N enriched) to nitrate provided by waters with less intense denitrification off California [31]. Likewise, mussels traced the spatial influence of the eutrophic vs. oligotrophic waters of the South African upwelling [30], and the same was reported for the upwelling gradient between Galicia and mar Cantábrico using δ¹⁵N measurements in *Fucus* sp. [48]. However, our study points out to the influence of additional sources, particularly during periods of low upwelling intensity. To reach the observed δ¹⁵N values in the coastal biomonitors studied, these sources must be enriched in ¹⁵N and most likely related to anthropogenic activity through the release of nitrate from waste waters,

either directly by discharges from rivers, rías, runoff, and groundwaters, or indirectly through the local remineralization of organic matter (including sewage) inside the rías. Even though only a few direct measurements of $\delta^{15}N$ signatures of dissolved nitrate are available in this area (see Table 3), the production of nitrate with high $\delta^{15}N$ can be attributed to the high nitrification rates as those measured in some Rías [70,81].

## Improvement of nitrogen biomonitoring

The techniques for using macroalgae and filter feeders as biomonitors of nutrient inputs are to be standardized. The complexity of accounting for the variability in $\delta^{15}N$ due to physiological processes, including seasonal growth or reproduction but also the turnover of different tissues, limited the applicability of both macroalgae [33,34] and mussels [79,82]. In this study we provide methods to minimize the influence of the major drivers of variability in the $\delta^{15}N$ signal, including the effect of seasonality and the absence of target biomonitor species in certain areas. Thus, seasonality in the $\delta^{15}N$ can be considered by adjusting the annual cycle observed at one representative location, as shown for both *F. spiralis* and *M. galloprovincialis* (S5 and S7 Figs) using equivalent empirical equations. Considering the similarities in seasonal cycles within the same region [51,83] the variability due to the differences in sampling time for intertidal organisms at different localities would be avoided. Similarly, comparisons of $\delta^{15}N$ between similar species would allow to extend monitoring time series beyond the locations where a particular species is present, as exemplified by *Fucus* sp. in the mar Cantábrico [53]. In addition, framing the sampling period according to the major inputs of new nutrients and focusing on tissues with known turnover would further reduce $\delta^{15}N$ variability.

Even though bulk $\delta^{15}N$ is relatively easy to monitor following the recommendations detailed above, the interpretation of the patterns found would be still difficult. One main reason is that bulk $\delta^{15}N$ represents the weighted average of $\delta^{15}N$ in all molecules present. The variety of compounds, each originated from different metabolic routes with characteristic isotopic fractionation, cause differences in compound-specific $\delta^{15}N$ even in primary producers [55]. Indeed, studies of biomagnification of pollutants using bulk $\delta^{15}N$ in mussels revealed high uncertainties in the resulting rates likely by the isotopic differences in the local nitrogen sources [84]. However, the consideration of $\delta^{15}N$ in some of these compounds, such as amino acids, allows for a better characterization of the signal from the nitrogen sources than bulk measurements do [31]. While it was not possible to reanalyze old mussel samples in our study to confirm patterns of change in amino acid $\delta^{15}N$, the similarity in the spatial pattern of $\delta^{15}N_{Phe}$ in mussels and $\Delta NO_3$ found for 2023 suggests that this marker is sensitive to changes in the proportion of upwelled nitrogen, and could be used to trace upwelling contributions in future monitoring studies. Additionally, selecting tissues with different turnover times will provide an index of the differential impact of the nitrogen sources in time. For example, using the digestive gland of mussels would provide estimations on nitrogen input at shorter time scales than those provided by the muscle [79].

## Conclusions

This study confirmed that upwelling remains the primary source of new nitrogen supporting coastal food webs in NW Spain. The occasional importance of other sources of nitrogen, likely of anthropogenic origin, and the absence of systematic trends in either upwelling or nutrient concentrations over the last 3 decades, suggest these additional sources compensate nitrogen inputs during periods of low upwelling intensity. This is evidenced by the observed increase in $\delta^{15}N$ of intertidal bioindicators, particularly in *Fucus* spp., with the decrease in the relative contribution of upwelling to nitrogen supply. The increase in $\delta^{15}N$ is consistent with a larger input of sources enriched in $^{15}N$, as nitrate derived from wastewaters and manure. Future research should focus on identifying and quantifying these sources, incorporating river inputs and runoff, either from time series measurements or from biogeochemical models. Finally, future monitoring assessments using $\delta^{15}N$ should be improved by targeting specific time periods (e.g., spring – early summer), and by considering more sensitive isotopic techniques, such as amino acid $\delta^{15}N$ in mussels.

 

## Supporting information

**S1 Fig. Upwelling index series.** Annual series of monthly averaged values for the upwelling index (UI, $m^3 s^{-1} km^{-1}$) for the three areas considered (see Fig 1). The continuous line indicates the center of gravity (weighted average of UI) of the main upwelling peak each year.
(ZIP)

**S2 Fig. Temperature, salinity and nitrate series.** Values of water temperature, salinity, and nitrate averaged from March to June in the surface (0–10 m) and bottom (70–100 m) layers for the three zones considered (see Fig 1). a: sea surface temperature (SST, °C), b: sea bottom temperature (SBT, °C), c: sea surface salinity (SSS), d: sea bottom salinity (SBS), e: surface nitrate ($SNO_3$, µM), f: bottom nitrate ($BNO_3$, µM). The black arrows indicate the tipping points of the upwelling index series (see Fig 2).
(ZIP)

**S3 Fig. $\delta^{15}N$ in _Fucus spiralis_ vs. _F. vesiculosus._** Box plot of $\delta^{15}N$ (‰) measured concurrently in _Fucus spiralis_ and _F. vesiculosus_ collected at the same location. Mean values do not differ significantly among species within the same sampling location (Student-t, $P > 0.05$). N = 6 for each species.
(ZIP)

**S4 Fig. $\delta^{15}N$ in _Fucus spp. vs. Bifurcaria bifurcata._** Correspondence between the $\delta^{15}N$ values (‰ measured in a) _Fucus serratus_ and either _F. vesiculosus_ (filled dots) or _F. spiralis_ (open dots), and b) _Bifurcaria bifurcata_ and _Fucus_ spp. sampled concurrently. Linear regression equations (± se), correlation coefficients (r), and significance (P) are indicated. Dots represent local mean ± sd values.
(ZIP)

**S5 Fig. Seasonality of $\delta^{15}N$ in _Fucus spiralis._** Seasonal variation of relative $\delta^{15}N$ anomalies (mean ± sd) from local minimum ($\Delta\delta^{15}N = \delta^{15}N_i - \delta^{15}N_{min}$) measured in _F. spiralis_ collected in Mera (see Fig 1). The line represents the adjusted cosine function ($P < 0.001$): $\Delta\delta^{15}N = C + \{a \cos [2 \pi (X-X_{min})/ T – p]\}$, where X is the Julian day and C (1.176), a (0.984), T (284.6), $X_{min}$ (21), and p (0.852) are regression parameters.
(ZIP)

**S6 Fig. Size variation of $\delta^{15}N$ in _Mytilus galloprovincialis._** Variation of $\delta^{15}N$ anomalies (mean ± sd, ‰) from local mean ($\delta^{15}N_i - \delta^{15}N_{mean}$) in 5 mm length size-classes of _Mytilus galloprovincialis._ The lower limit for each size-class (mm) is indicated in the horizontal axis labels. The dashed vertical line indicates the minimum size for size-independent anomalies.
(ZIP)

**S7 Fig. Seasonality of $\delta^{15}N$ in _Mytilus galloprovincialis._** Seasonal variation of relative mean (± sd) $\delta^{15}N$ ($\Delta\delta^{15}N = \delta^{15}N_i - \delta^{15}N_{min}$, ‰) measured in _Mytilus galloprovincialis_ in Mera (see Fig 1). The linear regression equation parameters (± se), correlation coefficients (r), and significance (P) are indicated.
(ZIP)

## Acknowledgments

We are indebted to all participants in the environmental time series project RADIALES and in the collection of samples for the Environmental Specimen Bank of Galicia. Particularly, we recognize the collaboration of J.R. Aboal, J.A. Fernández, and C. Pacín from the Universidade de Santiago de Compostela, as well as the dedicated work of C. Carballo, N. González, M. Castaño, L. Vázquez, and M. Álvarez, from IEO-CSIC, involved in the analysis of inorganic nutrients of the RADIALES series.

## Author contributions

**Conceptualization:** Antonio Bode, Rita García-Seoane, Zulema Varela, Inés G. Viana.

**Data curation:** Antonio Bode.

**Formal analysis:** Antonio Bode.

**Funding acquisition:** Antonio Bode.

**Investigation:** Antonio Bode, Rita García-Seoane, Inés G. Viana.

**Methodology:** Antonio Bode, Rita García-Seoane, Zulema Varela, Inés G. Viana.

**Resources:** Zulema Varela.

**Visualization:** Antonio Bode.

**Writing – original draft:** Antonio Bode.

**Writing – review & editing:** Antonio Bode, Rita García-Seoane, Zulema Varela, Inés G. Viana.

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
