## [Decision Letter · Decision Letter 0]

PONE-D-24-58553Assessment of decadal changes in coastal nitrogen sources in NW Spain with stable isotopes in macroalgae and musselsPLOS ONE

Dear Dr. Bode,

Thank you for submitting your manuscript to PLOS ONE. After careful consideration, we feel that it has merit but does not fully meet PLOS ONE’s publication criteria as it currently stands. Therefore, we invite you to submit a revised version of the manuscript that addresses the points raised during the review process.

We look forward to receiving your revised manuscript.

Kind regards,

Rajdeep Roy

Academic Editor

PLOS ONE

**Journal Requirements:**

1. When submitting your revision, we need you to address these additional requirements. Please ensure that your manuscript meets PLOS ONE's style requirements, including those for file naming. The PLOS ONE style templates can be found at https://journals.plos.org/plosone/s/file?id=wjVg/PLOSOne_formatting_sample_main_body.pdf and https://journals.plos.org/plosone/s/file?id=ba62/PLOSOne_formatting_sample_title_authors_affiliations.pdf 2. In your Methods section, please provide additional information regarding the permits you obtained for the work. Please ensure you have included the full name of the authority that approved the field site access and, if no permits were required, a brief statement explaining why. 3. Thank you for stating in your Funding Statement: This research was supported by MCIN/AEI/10.13039/501100011033 (Spain) through project QLOCKS (PID2020-115620RB-100), and additional funds by the Consellería de Medio Ambiente (Galicia, Spain) through the 3rd phase of the Environmental Specimen Bank (2000–2004), the Competitive Reference Group GRC GI-1252/GPC2020-23 (ED431C 2020/19), co-funded by the Xunta de Galicia (Spain) and the European Regional Development Fund (ERDF, EU), and postdoctoral research grants Juan de la Cierva-Incorporación to I.G.V. (IJC2019-040554-I), Juan de la Cierva-Formación to R.G.S. (FJC2019-040921-I), and María Zambrano Programme of the Spanish Ministry of Universities to Z.V. R.G.S also received funds from the European Union NextGenerationEU/PRTR programmes and is currently supported by the Horizon Europe research and innovation programme under a Marie Skłodowska-Curie Postdoctoral Fellowship 2023 (101150001-PelCon).  Please provide an amended statement that declares *all* the funding or sources of support (whether external or internal to your organization) received during this study, as detailed online in our guide for authors at http://journals.plos.org/plosone/s/submit-now.  Please also include the statement “There was no additional external funding received for this study.” in your updated Funding Statement. Please include your amended Funding Statement within your cover letter. We will change the online submission form on your behalf. 4. We noted in your submission details that a portion of your manuscript may have been presented or published elsewhere. Small subsets of the data on nitrogen stable isotopes in macroalgae were published in references 47, 49, 50, 56 and 57, as indicated in the manuscript. These data were used for different purposes of the objective of the present study. Please clarify whether this [conference proceeding or publication] was peer-reviewed and formally published. If this work was previously peer-reviewed and published, in the cover letter please provide the reason that this work does not constitute dual publication and should be included in the current manuscript. 5. When completing the data availability statement of the submission form, you indicated that you will make your data available on acceptance. We strongly recommend all authors decide on a data sharing plan before acceptance, as the process can be lengthy and hold up publication timelines. Please note that, though access restrictions are acceptable now, your entire data will need to be made freely accessible if your manuscript is accepted for publication. This policy applies to all data except where public deposition would breach compliance with the protocol approved by your research ethics board. If you are unable to adhere to our open data policy, please kindly revise your statement to explain your reasoning and we will seek the editor's input on an exemption. Please be assured that, once you have provided your new statement, the assessment of your exemption will not hold up the peer review process. 6. We note that you have referenced “Bode A, García-Seoane R, Varela Z, Viana IG” which has currently not yet been accepted for publication. Please remove this from your References and amend this to state in the body of your manuscript: (“Bode A, García-Seoane R, Varela Z, Viana IG”. [Submitted]) as detailed online in our guide for authorshttp://journals.plos.org/plosone/s/submission-guidelines#loc-reference-style 7. We note that Figure 1 in your submission contain map images which may be copyrighted. All PLOS content is published under the Creative Commons Attribution License (CC BY 4.0), which means that the manuscript, images, and Supporting Information files will be freely available online, and any third party is permitted to access, download, copy, distribute, and use these materials in any way, even commercially, with proper attribution. For these reasons, we cannot publish previously copyrighted maps or satellite images created using proprietary data, such as Google software (Google Maps, Street View, and Earth). For more information, see our copyright guidelines: http://journals.plos.org/plosone/s/licenses-and-copyright. We require you to either present written permission from the copyright holder to publish these figures specifically under the CC BY 4.0 license, or remove the figures from your submission: a. You may seek permission from the original copyright holder of Figure 1 to publish the content specifically under the CC BY 4.0 license.   We recommend that you contact the original copyright holder with the Content Permission Form (http://journals.plos.org/plosone/s/file?id=7c09/content-permission-form.pdf) and the following text:“I request permission for the open-access journal PLOS ONE to publish XXX under the Creative Commons Attribution License (CCAL) CC BY 4.0 (http://creativecommons.org/licenses/by/4.0/). Please be aware that this license allows unrestricted use and distribution, even commercially, by third parties. Please reply and provide explicit written permission to publish XXX under a CC BY license and complete the attached form.” Please upload the completed Content Permission Form or other proof of granted permissions as an "Other" file with your submission. In the figure caption of the copyrighted figure, please include the following text: “Reprinted from [ref] under a CC BY license, with permission from [name of publisher], original copyright [original copyright year].” b. If you are unable to obtain permission from the original copyright holder to publish these figures under the CC BY 4.0 license or if the copyright holder’s requirements are incompatible with the CC BY 4.0 license, please either i) remove the figure or ii) supply a replacement figure that complies with the CC BY 4.0 license. Please check copyright information on all replacement figures and update the figure caption with source information. If applicable, please specify in the figure caption text when a figure is similar but not identical to the original image and is therefore for illustrative purposes only.The following resources for replacing copyrighted map figures may be helpful: USGS National Map Viewer (public domain): http://viewer.nationalmap.gov/viewer/The Gateway to Astronaut Photography of Earth (public domain): http://eol.jsc.nasa.gov/sseop/clickmap/Maps at the CIA (public domain): https://www.cia.gov/library/publications/the-world-factbook/index.html and https://www.cia.gov/library/publications/cia-maps-publications/index.htmlNASA Earth Observatory (public domain): http://earthobservatory.nasa.gov/Landsat:
http://landsat.visibleearth.nasa.gov/USGS EROS (Earth Resources Observatory and Science (EROS) Center) (public domain): http://eros.usgs.gov/#Natural Earth (public domain): http://www.naturalearthdata.com/

**Additional Editor Comments:**

Comments to the authors

Since as a handling editor I had difficult times in getting your manuscript reviewed I took the liberty to do it my self being from the similar field. In addition to the comments received I have some more observations which I mention below. The authors are encouraged to go through.

The authors try to understand the relationships between the nitrogen uptake through N15 isotope (during the upwelling period) on macroalgae and the mussel. The data is collected over the years during the upwelling period.

My first question is both the experimental species are located in near shore/intertidal. One is a filter feeder and other I guess is attached to the substrata in very shallow waters. Whether the sampling location is affected by the upwelled waters.? How does authors prove this? This is not reflected by the salinity or temperature although some figures are given? The location is right next to shore so whether the hypothesis that upwelled might influence nitrate uptake is flawed? The authors need to justify. Why not change the orientation of the paper to “N15 isotope assimilation by intertidal species”?. Upwelling could be discussed as a paragraph in the discussion. Probably that’s the reason there is no change observed! Mussel responds strongly to calcium carbonate saturation and temperature fluctuation! The authors are encouraged to make a plot of showing the spatial extent of upwelling for any particular year to substantiate their scientific hypothesis. Also encouraged to plot the actual nitrate values from the location during monsoon and non-monsoon to show if any influence? A table showing a comparison of N15 assimilation by the (Mytilus galloprovincialis) from different other geographical area and for the macro algae can be added along with some discussion.

With the above suggestion the authors are encouraged to revise the present manuscript….

Reviewers' comments:

Reviewer's Responses to Questions

**Comments to the Author**

1. Is the manuscript technically sound, and do the data support the conclusions?

Reviewer #1: Yes

2. Has the statistical analysis been performed appropriately and rigorously? 

Reviewer #1: I Don't Know

3. Have the authors made all data underlying the findings in their manuscript fully available?

Reviewer #1: No

4. Is the manuscript presented in an intelligible fashion and written in standard English?

Reviewer #1: Yes

5. Review Comments to the Author

**Reviewer #1: ** The article being reviewed presents solid and well-structured research, with effective use of environmental data and isotopic analysis to assess changes in coastal nitrogen sources. The analysis covers a long period (1989–2023), allowing for the identification of possible long-term trends. However, some critical issues were noted regarding the description and clarity of the materials and methods section. Although the key concepts and main tools are described, essential details are missing, which may hinder the study’s reproducibility.

Several key issues were identified, including:

1. Lack of details on sampling and sample processing protocols.

o Rows 138–142: It is not clearly specified whether previous studies have used Bifurcaria bifurcata as a biomonitor and whether it could replace Fucus in this role.

o Row 151 mentions sampling a portion of the mussels adductor muscle but does not specify the exact quantity. Does this refer to a specific weight (grams) or length (centimetres), or is the quantity irrelevant?

2. Insufficient description of sample preservation. What measures were taken to prevent biological degradation after sampling?

3. Incomplete documentation of laboratory instruments and procedures.

o Row 121: It is recommended to specify exactly what the CTD is, including its model and parameters.

o Rows 151–153: The rationale behind the choice of drying or freeze-drying some samples is unclear. Was an existing protocol followed? If so, please specify which one. Otherwise, all steps leading to the preparation of samples for isotopic analysis should be described.

o Rows 142–143: If nitrogen isotopic measurements were extracted from the Environmental Specimen Bank of Galicia, it would be appropriate to specify exactly what type of information was selected. Were filters or other selection criteria used? Additionally, providing a link to this database would enhance transparency.

o Rows 149–150: It is not specified whether previous studies have observed correlations between isotopic signatures and shell size.

It is recommended that the materials and methods section be rewritten in a more comprehensive manner, as the current version lacks essential information needed to ensure reproducibility. If including all details in the main text would be overly cumbersome, they could be provided in tables, appendices, or supplementary materials, but they must be made available.

The quality of the figures is generally good, but some aspects could be improved to enhance readability and accessibility:

• Figure 1: Although latitude and longitude references are provided, the figure lacks a metric scale, which would aid in understanding the depicted distances. It is also suggested to use a colour other than black to indicate algae sampling areas, as this is difficult to distinguish from the land outline. Additionally, using a thicker symbol for Mytilus would improve visibility.

• To enhance accessibility, it is recommended to use a colour-blind-friendly palette, such as the one proposed by Wong, B. (2011). Points of View: Colour Blindness. Nat Methods, 8(441). https://doi.org/10.1038/nmeth.1618

It is advised that the authors carefully check the formatting of the article. For example:

• Row 738: 15N should be formatted as a superscript.

• Citations: References [19], [20], and [21] appear before [16], while [17] and [18] are missing. It is possible that in line 48, the citation indicated as [15, 19] should actually be [15–19].

6. PLOS authors have the option to publish the peer review history of their article (what does this mean? ). If published, this will include your full peer review and any attached files.

**Do you want your identity to be public for this peer review?** For information about this choice, including consent withdrawal, please see our Privacy Policy .

Reviewer #1: No

---

## [Author Response · Author response to Decision Letter 1]

7 May 2025

see response_letter attached after the revised manuscript

---

## [Editor Report · Decision Letter 1]

Assessment of decadal changes in coastal nitrogen sources in NW Spain with stable isotopes in macroalgae and mussels

PONE-D-24-58553R1

Dear Dr. Bode,

We’re pleased to inform you that your manuscript has been judged scientifically suitable for publication and will be formally accepted for publication once it meets all outstanding technical requirements.

Kind regards,

Rajdeep Roy

Academic Editor

PLOS ONE

Additional Editor Comments (optional):

Dear Dr. Bode

I am happy to say that you have made necessary corrections. The response to the questions raised by both the reviewers have been answered in a logical and scientific way. The manuscript has improved greatly. This is good contribution in the filed of oceanography.

Kind Regards

Academic Editor
---

## [Editor Report · Acceptance letter]

PONE-D-24-58553R1

PLOS ONE

Dear Dr. Bode,

I'm pleased to inform you that your manuscript has been deemed suitable for publication in PLOS ONE. Congratulations! Your manuscript is now being handed over to our production team.

Kind regards,

on behalf of

Dr. Rajdeep Roy

Academic Editor

PLOS ONE